# Impact of parameter control on the performance of APSO and PSO algorithms for the CSTHTS problem: An improvement in algorithmic structure and results

**Muhammad Ahmad Iqbal**[1], **Muhammad Salman Fakhar**[1]*, **Syed Abdul Rahman Kashif**[1‡], **Rehan Naeem**[1], **Akhtar Rasool**[2‡]

**1** Department of Electrical Engineering, University of Engineering and Technology, Lahore, Punjab, Pakistan, **2** Department of Electrical Engineering, Sharif College of Engineering and Technology, Lahore, Punjab, Pakistan

☯ These authors contributed equally to this work.
‡These authors also contributed equally to this work.
* salmanfakhar@uet.edu.pk

**Data Availability Statement:** All relevant data are within the paper.

## Abstract

Cascaded Short Term Hydro-Thermal Scheduling problem (CSTHTS) is a single objective, non-linear multi-modal or convex (depending upon the cost function of thermal generation) type of Short Term Hydro-Thermal Scheduling (STHTS), having complex hydel constraints. It has been solved by many metaheuristic optimization algorithms, as found in the literature. Recently, the authors have published the best-achieved results of the CSTHTS problem having quadratic fuel cost function of thermal generation using an improved variant of the Accelerated PSO (APSO) algorithm, as compared to the other previously implemented algorithms. This article discusses and presents further improvement in the results obtained by both improved variants of APSO and PSO algorithms, implemented on the CSTHTS problem.

## Introduction

The CSTHTS is a type of highly multi-modal, multi-dimensional, non-linear, and non-convex optimization problem which deals with the economic dispatch of power among hydel and thermal generating units. It can be modeled in generic form using the Eqs 1–9, as suggested in reference [1].

$$min(f) = \sum_{m=1}^{N} \sum_{i=1}^{T} n_m F_{(m,i)} \qquad (1)$$

where, $n_m$ is the number of hours in the scheduling period '$m$' and $F_{(m,i)}$ is the per hour cost of the thermal generating unit. Eq 1 presents the main objective function of the CSTHTS

**Funding:** The authors received no specific funding for this work.

**Competing interests:** The authors have declared that no competing interests exist.

**Abbreviations:** *APSO*, Accelerated particle swarm optimization; *CSTHTS*, Cascaded short-term hydrothermal scheduling; $F_{(m,i)}$, Fuel cost at $m^{th}$ scheduling interval of $i^{th}$ thermal generation unit; $I_{hyd_{j,m}}$, Inflow for the $j^{th}$ reservoir at $m^{th}$ scheduling interval; *N*, Total number of the scheduling interval; $N_h$, Total number of hydel generation units; $N_s$, Total number of thermal generation units; $P_d$, Load demand; $P_l$, Transmission losses; *PSO*, Particle swarm optimization; $P_{th_i}^{min}$, Minimum thermal power generated by the $i^{th}$ thermal generation unit.; $P_{th_{i,m}}$, Thermal power at $m^{th}$ scheduling interval of $i^{th}$ thermal generation unit; $P_{hyd_{j,m}}$, Hydel power at $m^{th}$ scheduling interval of $j^{th}$ thermal generation unit; $P_{th_i}^{max}$, Maximum thermal power generated by the $i^{th}$ thermal generation unit.; $P_{hyd_j}^{min}$, Minimum hydel power generated by the $j^{th}$ hydel generation unit; $P_{hyd_j}^{max}$, Maximum hydel power generated by the $j^{th}$ hydel generation unit; $Q_{hyd_{j,m}}$, The discharge rate of the $j^{th}$ reservoir at $m^{th}$ scheduling interval; $Q_{hyd_j}^{min}$, Minimum allowable value of the discharge rate of the $j^{th}$ reservoir based hydel generation unit at $m^{th}$ scheduling interval; $Q_{hyd_j}^{max}$, Maximum allowable value of the discharge rate of the $j^{th}$ reservoir based hydel generation unit at $m^{th}$ scheduling interval; $Q_{hyd_{j,total}}$, The total value of the allowed discharge rate of the $j^{th}$ reservoir based hydel generation unit for a total period of *N* scheduling interval; $R_{u,j}$, Number of the upstream reservoirs present for the $j^{th}$ reservoir; $S_{hyd_{j,m}}$, Spillage of the $j^{th}$ reservoir at $m^{th}$ scheduling interval; *t*, Water transport delay; $V_{hyd_{j,m}}$, Volume of the $j^{th}$ reservoir at $m^{th}$ scheduling interval; $V_{hyd_j}^{min}$, Minimum allowable value of the volume of the $j^{th}$ reservoir based hydel generation unit at $m^{th}$ scheduling interval; $V_{hyd_j}^{max}$, Maximum allowable value of the $j^{th}$ reservoir based hydel generation unit at $m^{th}$ scheduling interval.

problem, i.e., to minimize the total cost of the scheduling of thermal generators.

$$\sum_{m=1}^{N}\left(\sum_{i=1}^{N_s}P_{th_{i,m}} + \sum_{j=1}^{N_h}P_{hyd_{j,m}}\right) = P_d + P_l \tag{2}$$

The equality constraint of the CSTHTS problem is shown by Eq 2, which assures that the sum of powers generated by both thermal $P_{th_{i,m}}$ and hydel $P_{hyd_{j,m}}$ plants are equal to the sum of demand power $P_d$ and transmission losses $P_l$ of the power system for a total time of *N* scheduling intervals. Where, $N_s$ is the total number of thermal plants, $N_h$ is the total number of water reservoirs.

$$P_{hyd_{j,m}} = f\left(V_{hyd_{j,m}}, Q_{hyd_{j,m}}\right) \tag{3}$$

Eq 3 presents hydel power as a function of the volume $V_{hyd_{j,m}}$ and water discharge rate $Q_{hyd_{j,m}}$ at scheduling interval '$m$' of $j^{th}$ reservoir.

$$P_{th_i}^{min} \leq P_{th_{i,m}} \leq P_{th_i}^{max} \tag{4}$$

$$P_{hyd_j}^{min} \leq P_{hyd_{j,m}} \leq P_{hyd_j}^{max} \tag{5}$$

Eqs 4 and 5 give the bounded power limits for the $i^{th}$ thermal and the $j^{th}$ hydel unit, respectively at the $m^{th}$ scheduling interval. Where, $P_{th_i}^{min}$ and $P_{th_i}^{max}$ are the minimum and maximum allowable value of thermal generation for the $i^{th}$ thermal unit, respectively. And $P_{hyd_j}^{min}$ and $P_{hyd_j}^{max}$ are the minimum and maximum allowable value of hydel generation for the $j^{th}$ water reservoir, respectively.

$$V_{hyd_j}^{min} \leq V_{hyd_{j,m}} \leq V_{hyd_j}^{max} \tag{6}$$

$$Q_{hyd_j}^{min} \leq Q_{hyd_{j,m}} \leq Q_{hyd_j}^{max} \tag{7}$$

Eqs 6 and 7 are related to the operation of the water reservoir. Where, $V_{hyd_j}^{min}$ and $V_{hyd_j}^{max}$ are the minimum and maximum allowable value of the volume of the $j^{th}$ reservoir at $m^{th}$ scheduling interval and $Q_{hyd_j}^{min}$ and $Q_{hyd_j}^{max}$ are the minimum and maximum allowable value of the discharge rate of the $j^{th}$ reservoir at $m^{th}$ scheduling interval.

$$\sum_{m=1}^{N}Q_{hyd_{j,m}} = Q_{hyd_{j,total}} \tag{8}$$

Eq 8 gives the allowed value of the water discharged $Q_{hyd_{j,total}}$ by the $j^{th}$ reservoir for a total time of *N* scheduling intervals.

$$V_{hyd_{j,(m+1)}} = \quad V_{hyd_{j,m}} + I_{hyd_{j,m}} - Q_{hyd_{j,m}} - S_{hyd_{j,m}} +$$
$$\sum_{a=1}^{R_{u,j}}\left(Q_{hyd_{a,(m-t)}} + S_{hyd_{a,(m-t)}}\right) \tag{9}$$

The continuity equation balances the discharges and reservoir volume and is shown by Eq 9. Where, '$m$' is the scheduling hour, '$j$' is the reservoir available for hydel generation, $I_{hyd_{j,m}}$ is the inflow of the reservoir, $S_{hyd_{a,(m-t)}}$ is the spillage of the reservoir, $R_{u,j}$ are the upstream

reservoirs present for the $j^{th}$ reservoir and '$t$' is the water transport delay from reservoir '$a$' to reservoir '$j$'.

$$F_{(m,i)} = a + bP_{th_{(i,m)}} + cP^2_{th_{(i,m)}} \qquad (10)$$

The cost $F_{(m,i)}$ of $i^{th}$ thermal unit at $m^{th}$ scheduling interval is the function of the power of the thermal generator which is the function of the fuel cost. The relation of cost and thermal power is given by Eq 10, which is a quadratic function (convex) of thermal power and '$a$', '$b$' and '$c$' are the coefficients of the scheduling equation.

Eq 10 can be of higher order depending upon the type of thermal generator, causing an increase in the nonlinearity of the objective function. The CSTHTS problem is a scheduling problem having many scheduling intervals and each interval is treated as one dimension of the problem, which makes the CSTHTS highly multi-modal, i.e., a problem with an objective function having multiple peaks.

The test system of the CSTHTS problem selected in this article is a twenty-four hours long scheduling problem having one thermal unit and four hydel units for power generation. It is solved without considering the valve-point effect of thermal units and transmission losses. Data for the cost coefficients of the thermal unit and power production coefficients of hydro units are taken from reference [2].

It can be misinterpreted as a convex problem by just looking at the thermal fuel cost function (which is quadratic) as it is the main objective function but as a whole when this objective function is taken along with all the time coupling hydel and thermal constraints for twenty-four scheduling intervals it becomes multi-modal and non-convex because the thermal power generated depends upon the hydel generation, as mentioned in Eq 2. The function used to calculate hydel generation, as mentioned in Eq 3 is non-convex in nature as it contains multiplication terms of volume and discharge rate of the reservoir generating power to meet the load demand. The multiplication of two variables is a non-convex function according to reference [3]. This article discusses a highly complex, multi-modal, and non-convex case of the CSTHTS problem having quadratic thermal fuel cost function, benchmark case of the CSTHTS problem.

The CSTHTS problem, according to reference [1], is concerned with the combined economic dispatch of a chain of numerous water reservoirs located on the same stream in sequence, i.e., one reservoir-based hydel power plant is downhill from the other reservoir-based hydel power plant. There may be numerous thermal power plants in such problems, although they are normally treated as equivalent thermal generating units. if $j > 1$, the CSTHTS problem is defined by equality and non-equality equations given in Eqs 1–9.

The genetic algorithm has been implemented in reference [2] to optimize the CSTHTS problem. Reference [4], has implemented the dynamic search space squeezing based APSO on the STHTS problem. Reference [5], presents APSO and firefly algorithm on the hybrid case of STHTS problem which incorporates the effects of adding solar cells to the conventional grid. Dynamically search space squeezing based firefly algorithm has been implemented on the hybrid case of the STHTS problem in reference [6]. References [4–6], have shown that the superiority of one metaheuristic algorithm over the other should be made by performing the proper statistical tests, not by just comparing their achieved optimal solution. Three variants of evolutionary programming algorithms have been implemented to solve the CSTHTS problem, known as FEP, IFEP, and CEP, in reference [7]. The particle swarm optimization (PSO) algorithm has been implemented in reference [8] to solve the CSTHTS problem. The CSTHTS problem has been solved by adaptive and modified adaptive PSO in reference [9]. The differential evolution algorithm has been implemented in reference [10] to optimize the CSTHTS

problem. The real coded genetic algorithm has been used in reference [11] to find the optimal solution to the CSTHTS problem. The IPSO, EPSO, and EGA algorithms have been implemented in reference [12] to optimize the CSTHTS problem. The modified differential evolution algorithm has been used in reference [13] to solve the CSTHTS problem. The teaching learning-based algorithm has been used in reference [14] to find the approximate global optimum solution of the CSTHTS problem. Reference [15], has used an artificial fish swarm and a real coded genetic hybrid algorithm to solve the CSTHTS problem. Reference [16], implemented PSO-ALNS algorithm, moth-flame optimization, grey wolf optimization, and a combination of grey wolf and dragonfly algorithm to obtain the global optimum solution of the CSTHTS problem. Reference [17], showed that the small population-based PSO to be the best algorithm to find the optimal robust solution of the CSTHTS problem.

A detailed review of the CSTHTS problem solved by different conventional and metaheuristic algorithms in past was presented in reference [18]. The improved PSO with adaptive cognitive and social components has been implemented in reference [19], to optimize the CSTHTS problem. In Reference [19], the algorithm has two update equations which increase the complexity of implementing it on large-scale optimization problems. A novel PSO technique has been implemented in reference [20], which used the concept of adaptive inertia weight constant to solve the CSTHTS problem. In this technique, the social and cognitive coefficients were not considered. The diversified PSO technique has been implemented in reference [21], to find the approximate global optimum solution of the CSTHTS problem. However, this suggested technique depends upon the optimal selection of the population size.

PSO and APSO in their canonical versions can have an issue of premature convergence. One of the most recent methods to cope with the issue of premature convergence is to dynamically utilize extreme learning machine mechanisms along with adopting mutation strategy to have a dynamic update of the particles, as presented and proved by reference [22]. However, it has been carefully observed by the authors of this article that adopting mutation strategies along with extreme learning machines is not required in the newly proposed variants of improved APSO and improved PSO, while adopting the newly suggested constraint handling technique. The tuning of $\alpha$ and $\beta$ coefficients in APSO variants and the acceleration and inertia coefficients in improved PSO makes sure that the particles do not get stuck to local optima, by improving the global search ability of the particles at the initial stages of the iterative process. However, for extremely high multi-modal and non-convex optimization problems, premature convergences can be avoided by utilizing the techniques suggested by reference [22].

According to no free lunch theorems presented in reference [23], if an algorithm performs well when applied to an optimization problem, it does not guarantee that it will perform well on other types of optimization problems. Therefore, it is required to find a suitable optimization algorithm for each type of optimization problem, if taken individually. In this article, a highly complex and multi-modal case of the CSTHTS problem is solved by implementing sixteen variants of the APSO algorithm and one variant of the PSO algorithm. The results obtained have been compared with previously found optimal solutions of the selected test case by other conventional and metaheuristic algorithms, reported in the literature.

In this article, the CSTHTS case has been solved by using the newly proposed sixteen improved variants of APSO and an improved variant of PSO along with a newly proposed constraint handling technique as an original contribution. It has been presented that the PSO variant has given so far the best results of the CSTHTS case, as compared to the previously reported results of the implementation of other algorithms, as found in literature [24]. Reference [24], previously published by the authors of this article, gave the so far best-achieved results of the selected test case of the CSTHTS problem. However, there were two motivations to further improve the algorithms in search of finding better solutions. Firstly, the results were

achieved in reference [24] used a larger number of particles (5000) and a large number of iterations (5000), having a high value of standard deviation, and the average time took to simulate one trial was 243 seconds. Therefore, it was needed to further improve the algorithms so that the results can be achieved using a smaller number of particles, having a lower value of standard deviation, and in lesser computational time. Secondly, it was needed to further investigate that if using the new modifications in APSO and PSO algorithms can help to achieve further improvement in results. The findings of this new experiment on the CSTHTS case by designing and implementing the newly proposed variants of PSO and APSO have been presented in this article. It has been shown that both improved variants of APSO and PSO helped to achieve better results in a very less computational time as compared to reference [24]. Though the improved PSO was able to find the nearest approximation to the global optimum solution, however, for repeated trials, the standard deviation of the results achieved by the improved PSO variant remained very high. The sixteen improved variants of APSO then served the purpose of keeping the standard deviation value lower while achieving almost near results to that achieved by the improved PSO variant for the best trial. A comparative analysis of the results achieved by the improved PSO variant is made with the best among the sixteen variants of improved APSO using statistical tests and is presented in this article.

The paper is organized as follows: Section 2 describes the algorithmic structure of the sixteen variants of APSO along with presenting the pseudo-code for the newly proposed constraint handling technique for the CSTHTS case, Section 3 presents the algorithmic structure of the improved PSO variant, Section 4 describes the results of the implementations of improved PSO variant and sixteen variants of improved APSO on the CSTHTS case, Section 5 gives the discussion on the results achieved, and finally, section 6 concludes the paper.

## Improvements of APSO algorithm

The APSO algorithm, presented in reference [23], is a very promising version of the canonical PSO. When compared to the original PSO algorithm, it can be witnessed that it has a single update equation as described by Eq 11 for the particles without requiring the velocity update equation, making it simple to apply to the optimization problem.

$$x_i^{t+1} = (1 - \beta)x_i^t + \beta g + \alpha \epsilon \tag{11}$$

where, the typical values of $\alpha$ and $\beta$ are usually taken as 0.2 and 0.5 respectively in the canonical version.

In this article, the authors have applied another modification in the improved APSO algorithm, that was implemented on the CSTHTS case in reference [24]. The improved variant has a single-step update equation without utilizing the velocity update equation as given in Eq 12.

$$x_i^{t+1} = (1 - \beta(t))p_i^t + \beta(t)g^t + \alpha(t)R_i^t \tag{12}$$

The values of $\alpha$ and $\beta$ should be in between 0 and 1 and their most effective range of performance is in between the maximum and minimum limits mentioned in reference [23]. Increasing the values of $\alpha$ and $\beta$ greater than 1 causes the solution of the APSO algorithm to diverge. The values of $\alpha$ and $\beta$ are normally taken as fixed in the canonical form, according to reference [23]. The improvements are made in the canonical APSO by modifying the $\alpha$ and $\beta$ coefficients. In reference [24], $\alpha$ is varied from 0.6 to 0.2 and $\beta$ is varied from 0.7 to 0.1. The authors of this paper add further improvement in the improved version of the APSO algorithm of reference [24] and suggest sixteen different variants of improved APSO in terms of $\alpha$ and $\beta$ parameter control. The results achieved are significantly better than the results of reference

[24]. Another improvement in Eq 11, is using the local best $p_i^t$ of each particle at any iteration $k$, instead of the particle's current position at time $t$, as suggested in reference [25].

Where, $R_i^t$ is an $N \times d$ dimensional matrix of uniform random numbers given in reference [25], where $N$ is the particles generated and $d$ is the total scheduling intervals of the CSTHTS problem. Randomization of the APSO algorithm is increased by using higher values of $\alpha$ to avoid premature convergence to local optima. When the APSO algorithm progresses to the end of iterations, during this time particles are converging towards the global optimum solution avoiding the local optimum solutions. So, it is required that the particles do not diverge (oscillates) from the global optimum solution. Therefore, $\alpha$ is decreased from a higher value to a lower value as the algorithm proceeds from the first iteration to the last iteration. This decay in the value of $\alpha$ may be sinusoidal, linear, quadratic, or exponential depending upon the type of optimization problem under consideration. It was observed that the linear decay in the value of $\alpha$ gives a good performance of the APSO algorithm applied to the CSTHTS problem.

The value of $\beta$ decides the weight given to the particle global best and local best of the iteration. It can be a constant as well as varying value. It has been observed that different chaotic maps applied to vary the value of $\beta$ gives a good performance of the APSO algorithm instead of making it fixed. This will help to keep the solution space diversity alive, similar to the case of the levy flight concept in cuckoo search and firefly algorithm.

To conclude, at the starting point of the APSO algorithm, the $\alpha$ value should be high to increase the particles exploration. However, step reduction of the value of $\alpha$ should be made as the APSO algorithm start proceeding towards the end to make the APSO algorithm converge at a good approximate of the global optimum solution. Higher values of $\beta$ allow particles to have a greater influence from the global best of each iteration and the lower value to have more influence from the local best of each iteration. Brilliant results have been achieved by making these modifications in the canonical APSO algorithm.

## Variants of APSO

Parameter control setting of APSO for sixteen different chaotic maps has been implemented on the test case of the CSTHTS problem, which gives improved results compared with reference [24], among all the implemented chaotic maps, variant-16 gives the minimum cost. It is important to mention that a newly proposed constraint handling technique has been incorporated by the authors while implementing all the variants of APSO. The pseudo-code of this constraint handling technique has been presented in Table 1. Each variant is simulated for 50

**Table 1. Pseudocode for newly proposed constraint handling technique.**

| |
|---|
| **If** (Qi,d <Qmin) |
| Find difference = Qmin—Qi,d |
| Set Qi,d = Qmin (hard limit) |
| **End If** |
| **For** n = 23 to 1 |
| Adjust the difference in 23 entries while keeping them in limits. Starting from 23rd entry to 1st entry. |
| **End For** |
| **Else If** (Qi,d >Qmax) |
| Find difference = Qi,d– Qmax |
| Set Qi,d = Qmax (hard limit) |
| **For** n = 1 to 23 |
| Adjust the difference in 23 entries while keeping them in limits. Starting from 1st entry to 23rd entry. |
| **End For** |

trials having 5050 iterations and population size of 600. Details of sixteen chaotic maps implemented are given as follows.

**Variant 1.** In this variant, $\alpha$ is sinusoidally decreased up to 1/4 of wavelength (90°) from 0.81 to 0.62. Values of $\alpha_{max}$ and $\alpha_{min}$ in this variant are 0.81 and 0.62, respectively as shown in Eq 13. The value of $\beta$ is linearly decreased from 0.81 to 0.62. Values of $\beta_{max}$ and $\beta_{min}$ in this variant are 0.81 and 0.62, respectively as shown in Eq 14.

$$\alpha = \alpha_{min} + (\alpha_{max} - \alpha_{min}) \times \cos\left(\frac{\pi}{2} \times \frac{Iteration_{current}}{Iteration_{total}}\right) \tag{13}$$

$$\beta = \beta_{max} - \left((\beta_{max} - \beta_{min}) \times \frac{Iteration_{current}}{Iteration_{total}}\right) \tag{14}$$

**Variant 2.** In this variant, $\alpha$ is linearly decreased from 0.81 to 0.62. Values of $\alpha_{max}$ and $\alpha_{min}$ in this variant are 0.81 and 0.62, respectively as shown in Eq 15. The value of $\beta$ is sinusoidally decreased up to 1/2 of wavelength (180°) from 0.81 to 0.62. Values of $\beta_{max}$ and $\beta_{min}$ in this variant are 0.81 and 0.715, respectively as shown in Eq 16.

$$\alpha = \alpha_{max} - \left((\alpha_{max} - \alpha_{min}) \times \frac{Iteration_{current}}{Iteration_{total}}\right) \tag{15}$$

$$\beta = \beta_{min} + (\beta_{max} - \beta_{min}) \times \cos\left(\pi \times \frac{Iteration_{current}}{Iteration_{total}}\right) \tag{16}$$

**Variant 3.** In this variant, $\alpha$ and $\beta$ both are sinusoidally decreased up to 1/4 of wavelength (90°) from 0.81 to 0.62. Values of $\alpha_{max}$, $\alpha_{min}$, $\beta_{max}$ and $\beta_{min}$ in this variant are 0.81, 0.62, 0.81 and 0.62, respectively as shown in Eqs 17 and 18.

$$\alpha = \alpha_{min} + (\alpha_{max} - \alpha_{min}) \times \cos\left(\frac{\pi}{2} \times \frac{Iteration_{current}}{Iteration_{total}}\right) \tag{17}$$

$$\beta = \beta_{min} + (\beta_{max} - \beta_{min}) \times \cos\left(\frac{\pi}{2} \times \frac{Iteration_{current}}{Iteration_{total}}\right) \tag{18}$$

**Variant 4.** In this variant, $\alpha$ and $\beta$ are linearly decreased from 0.81 to 0.62. Values of $\alpha_{max}$, $\alpha_{min}$, $\beta_{max}$ and $\beta_{min}$ in this variant are 0.81, 0.62, 0.81 and 0.62, respectively as shown in Eqs 19 and 20.

$$\alpha = \alpha_{max} - \left((\alpha_{max} - \alpha_{min}) \times \frac{Iteration_{current}}{Iteration_{total}}\right) \tag{19}$$

$$\beta = \beta_{max} - \left((\beta_{max} - \beta_{min}) \times \frac{Iteration_{current}}{Iteration_{total}}\right) \tag{20}$$

**Variant 5.** In this variant, $\alpha$ and $\beta$ both are sinusoidally decreased up to 1/2 of wavelength (180°) from 0.81 to 0.62. Values of $\alpha_{max}$, $\alpha_{min}$, $\beta_{max}$ and $\beta_{min}$ in this variant are 0.81, 0.715, 0.81

and 0.715, respectively as shown in Eqs 21 and 22.

$$\alpha = \alpha_{min} + (\alpha_{max} - \alpha_{min}) \times \cos\left(\pi \times \frac{Iteration_{current}}{Iteration_{total}}\right) \tag{21}$$

$$\beta = \beta_{min} + (\beta_{max} - \beta_{min}) \times \cos\left(\pi \times \frac{Iteration_{current}}{Iteration_{total}}\right) \tag{22}$$

**Variant 6.**   In this variant, $\alpha$ is sinusoidally decreased up to 1/2 of wavelength (180˚) from 0.81 to 0.62. Values of $\alpha_{max}$ and $\alpha_{min}$ in this variant are 0.81 and 0.715, respectively as shown in Eq 23. The value of $\beta$ is linearly decreased from 0.81 to 0.62. Values of $\beta_{max}$ and $\beta_{min}$ in this variant are 0.81 and 0.62, respectively as shown in Eq 24.

$$\alpha = \alpha_{min} + (\alpha_{max} - \alpha_{min}) \times \cos\left(\pi \times \frac{Iteration_{current}}{Iteration_{total}}\right) \tag{23}$$

$$\beta = \beta_{max} - \left((\beta_{max} - \beta_{min}) \times \frac{Iteration_{current}}{Iteration_{total}}\right) \tag{24}$$

**Variant 7.**   In this variant, $\alpha$ is sinusoidally decreased up to 1/4 of wavelength (90˚) from 0.81 to 0.62. Values of $\alpha_{max}$ and $\alpha_{min}$ in this variant are 0.81 and 0.62, respectively as shown in Eq 25. The value of $\beta$ is fixed at 0.715.

$$\alpha = \alpha_{min} + (\alpha_{max} - \alpha_{min}) \times \cos\left(\frac{\pi}{2} \times \frac{Iteration_{current}}{Iteration_{total}}\right) \tag{25}$$

**Variant 8.**   In this variant, $\alpha$ is linearly decreased from 0.81 to 0.62. Values of $\alpha_{max}$ and $\alpha_{min}$ in this variant are 0.81 and 0.62, respectively as shown in Eq 26. The value of $\beta$ is fixed at 0.715.

$$\alpha = \alpha_{max} - \left((\alpha_{max} - \alpha_{min}) \times \frac{Iteration_{current}}{Iteration_{total}}\right) \tag{26}$$

**Variant 9.**   In this variant, $\alpha$ is sinusoidally decreased up to 1/2 of wavelength (180˚) from 0.81 to 0.62. Values of $\alpha_{max}$ and $\alpha_{min}$ in this variant are 0.81 and 0.715, respectively as shown in Eq 27. The value of $\beta$ is fixed at 0.715.

$$\alpha = \alpha_{min} + (\alpha_{max} - \alpha_{min}) \times \cos\left(\pi \times \frac{Iteration_{current}}{Iteration_{total}}\right) \tag{27}$$

**Variant 10.**   In this variant, $\alpha$ is sinusoidally decreased up to 1/4 of wavelength (90˚) from 0.81 to 0.62. Values of $\alpha_{max}$ and $\alpha_{min}$ in this variant are 0.81 and 0.62, respectively as shown in Eq 28. The value of $\beta$ is linearly increased from 0.62 to 0.81. Values of $\beta_{max}$ and $\beta_{min}$ in this

variant are 0.81 and 0.62, respectively as shown in Eq 29.

$$\alpha = \alpha_{min} + (\alpha_{max} - \alpha_{min}) \times \cos\left(\frac{\pi}{2} \times \frac{Iteration_{current}}{Iteration_{total}}\right) \qquad (28)$$

$$\beta = \beta_{min} + \left((\beta_{max} - \beta_{min}) \times \frac{Iteration_{current}}{Iteration_{total}}\right) \qquad (29)$$

**Variant 11.** In this variant, $\alpha$ is sinusoidally decreased up to 1/2 of wavelength (180˚) from 0.81 to 0.62. Values of $\alpha_{max}$ and $\alpha_{min}$ in this variant are 0.81 and 0.715, respectively as shown in Eq 30. The value of $\beta$ is linearly increased from 0.62 to 0.81. Values of $\beta_{max}$ and $\beta_{min}$ in this variant are 0.81 and 0.62, respectively as shown in Eq 31.

$$\alpha = \alpha_{min} + (\alpha_{max} - \alpha_{min}) \times \cos\left(\pi \times \frac{Iteration_{current}}{Iteration_{total}}\right) \qquad (30)$$

$$\beta = \beta_{min} + \left((\beta_{max} - \beta_{min}) \times \frac{Iteration_{current}}{Iteration_{total}}\right) \qquad (31)$$

**Variant 12.** In this variant, $\alpha$ is linearly decreased from 0.81 to 0.62. Values of $\alpha_{max}$ and $\alpha_{min}$ in this variant are 0.81 and 0.62, respectively as shown in Eq 32. The value of $\beta$ is sinusoidally decreased up to 1/4 of wavelength (90˚) from 0.81 to 0.62. Values of $\beta_{max}$ and $\beta_{min}$ in this variant are 0.81 and 0.62, respectively as shown in Eq 33.

$$\alpha = \alpha_{max} - \left((\alpha_{max} - \alpha_{min}) \times \frac{Iteration_{current}}{Iteration_{total}}\right) \qquad (32)$$

$$\beta = \beta_{min} + (\beta_{max} - \beta_{min}) \times \cos\left(\frac{\pi}{2} \times \frac{Iteration_{current}}{Iteration_{total}}\right) \qquad (33)$$

**Variant 13.** In this variant, $\alpha$ is sinusoidally decreased up to 1/2 of wavelength (180˚) from 0.81 to 0.62. Values of $\alpha_{max}$ and $\alpha_{min}$ in this variant are 0.81 and 0.715, respectively as shown in Eq 34. The value of $\beta$ is sinusoidally increased up to 1/2 of wavelength (180˚) from 0.62 to 0.81. Values of $\beta_{max}$ and $\beta_{min}$ in this variant are 0.81 and 0.715, respectively as shown in Eq 35.

$$\alpha = \alpha_{min} + (\alpha_{max} - \alpha_{min}) \times \cos\left(\pi \times \frac{Iteration_{current}}{Iteration_{total}}\right) \qquad (34)$$

$$\beta = \beta_{min} - (\beta_{max} - \beta_{min}) \times \cos\left(\pi \times \frac{Iteration_{current}}{Iteration_{total}}\right) \qquad (35)$$

**Variant 14.** In this variant, $\alpha$ is linearly decreased from 0.81 to 0.62. Values of $\alpha_{max}$ and $\alpha_{min}$ in this variant are 0.81 and 0.62, respectively as shown in Eq 36. The value of $\beta$ is sinusoidally increased up to 1/2 of wavelength (180˚) from 0.81 to 0.62. Values of $\beta_{max}$ and $\beta_{min}$ in

this variant are 0.81 and 0.715, respectively as shown in Eq 37.

$$\alpha = \alpha_{max} - \left( (\alpha_{max} - \alpha_{min}) \times \frac{Iteration_{current}}{Iteration_{total}} \right) \tag{36}$$

$$\beta = \beta_{min} - (\beta_{max} - \beta_{min}) \times \cos\left( \pi \times \frac{Iteration_{current}}{Iteration_{total}} \right) \tag{37}$$

**Variant 15.** In this variant, $\alpha$ is sinusoidally decreased up to 1/4 of wavelength (90˚) from 0.81 to 0.62. Values of $\alpha_{max}$ and $\alpha_{min}$ in this variant are 0.81 and 0.62, respectively as shown in Eq 38. The value of $\beta$ is sinusoidally increased up to 1/4 of wavelength (90˚) from 0.62 to 0.81. Values of $\beta_{max}$ and $\beta_{min}$ in this variant are 0.81 and 0.62, respectively as shown in Eq 39.

$$\alpha = \alpha_{min} + (\alpha_{max} - \alpha_{min}) \times \cos\left( \frac{\pi}{2} \times \frac{Iteration_{current}}{Iteration_{total}} \right) \tag{38}$$

$$\beta = \beta_{min} + (\beta_{max} - \beta_{min}) \times \sin\left( \frac{\pi}{2} \times \frac{Iteration_{current}}{Iteration_{total}} \right) \tag{39}$$

**Variant 16.** In this variant, $\alpha$ is linearly decreased from 0.81 to 0.62. Values of $\alpha_{max}$ and $\alpha_{min}$ in this variant are 0.81 and 0.62, respectively as shown in Eq 40. The value of $\beta$ is sinusoidally increased up to 1/4 of wavelength (90˚) from 0.81 to 0.62. Values of $\beta_{max}$ and $\beta_{min}$ in this variant are 0.81 and 0.62, respectively as shown in Eq 41. This gives the same variant as was utilized in reference [24], however, the minimum and maximum values of tuning parameters $\alpha$ and $\beta$ play a significant role in improving the results for the CSTHTS problem. Moreover, by using these values of $\alpha$ and $\beta$, the required number of particles is reduced to 600, as compared to 5000 particles used in reference [24], which is a significant computational improvement. The computational time of simulating one trial of this variant is 45 seconds.

$$\alpha = \alpha_{max} - \left( (\alpha_{max} - \alpha_{min}) \times \frac{Iteration_{current}}{Iteration_{total}} \right) \tag{40}$$

$$\beta = \beta_{min} + (\beta_{max} - \beta_{min}) \times \sin\left( \frac{\pi}{2} \times \frac{Iteration_{current}}{Iteration_{total}} \right) \tag{41}$$

## Improved PSO algorithm

The particle swarm optimization (PSO) in its canonical form has been a brilliant algorithm used for solving several optimization problems as found in the literature, according to [18]. However, the PSO algorithm in its canonical form was implemented in reference [26] on the CSTHTS case, and the results reported were under-rated. It has been observed that the constraint handling technique used in the reference [26] was the penalty factor approach. By using the newly proposed constraint handling technique of Table 1, PSO achieves markedly improved results.

The PSO has now been a very well-known and promising optimization algorithm in metaheuristic algorithms. It has been applied to many non-linear, complex, and multi-modal optimization problems to give brilliant results. There are almost more than two dozen of its

variants as discussed in reference [23]. The PSO has two equations as shown in Eqs 42 and 43.

$$v_i^{n+1} = wv_i^n + (\alpha(n)\epsilon_1 \times (g^{*n} - x_i^n)) + (\beta(n)\epsilon_2 \times (p_i^n - x_i^n)) \tag{42}$$

$$x_i^{n+1} = x_i^n + v_i^{n+1} \tag{43}$$

where, $v_i^{n+1}$ is known as velocity, in PSO terminology, that is added in current position $x_i^n$ of the particle to get the new position $x_i^{n+1}$ and $g$ is the global best and $p$ is the local best of $i^{th}$ particle of $n^{th}$ iteration. The acceleration coefficients are $\alpha$ and $\beta$, usually varied between 0 to 2.1, $w$ is the inertial factor of the particles having a value in the range of 0 to 1, and $\epsilon$ is a random number generated in between 0 and 1, according to reference [23]. Lower values of acceleration coefficients make the particle's trajectory smooth while higher values cause abrupt movements. The inertial factor controls the impact of the previous velocity in a new direction and is used to balance exploration and exploitation. Larger values of inertial factor result in exploration (may cause divergence of the swarm) and small values cause exploitation (decelerate the particles resulting in the convergence of swarm). The values of the tuning parameters of the PSO update equation can be taken as constant as well as varying per the number of iterations (parameter control).

It has been observed solving the CSTHTS problem that the constant values of these tuning parameters of PSO do not give good results but using the parameter control for the tuning parameters, PSO gives brilliant results. Value of $\alpha$ and $w$ is linearly decreased from $\alpha_{max}$ = 1.95 to $\alpha_{min}$ = 1.8 and $w_{max}$ = 0.001 to $w_{min}$ = 0, respectively and $\beta$ value is sinusoidally decreased from $\beta_{max}$ = 1.95 to $\beta_{min}$ = 1.8 $\left(\frac{1}{2} \text{ to } \frac{3}{4} \text{ of wavelength}\right)$ in this research as shown in Eqs 44–46. This variant of PSO is simulated for 50 trials having 2500 iterations and population size of 500. The computational time of simulating one trial of this variant is 34 seconds.

$$w = w_{max} - \left((w_{max} - w_{min}) \times \frac{Iteration_{current}}{Iteration_{total}}\right) \tag{44}$$

$$\alpha = \alpha_{max} - \left((\alpha_{max} - \alpha_{min}) \times \frac{Iteration_{current}}{Iteration_{total}}\right) \tag{45}$$

$$\beta = \beta_{max} - \left((\beta_{max} - \beta_{min}) \times \sin\left(\frac{\pi}{2} \times \frac{Iteration_{current}}{Iteration_{total}}\right)\right) \tag{46}$$

## Results

The test system selected in this article is simulated on MATLAB run on Intel(R) Core (TM) i3-3220 CPU at 3.30 GHz system having 8 GB RAM. The minimum cost achieved by the sixteen variants of the APSO algorithm is shown in Table 2. It can be seen that APSO variant-16 gives the best results compared with the remaining fifteen variants of APSO in terms of minimum, maximum, average, and standard deviation of cost achieved for the CSTHTS problem, considered in this article. Fig 1 shows the convergence behavior of APSO variant-16 applied to the CSTHTS problem.

The PSO variant (Improved PSO) applied to the CSTHTS problem gives the best-achieved minimum result so far compared to the sixteen variants of APSO discussed in this article as well as results reported in the literature, according to the reference [24]. However, the maximum, average, and standard deviation of the cost obtained by implementing Improved PSO

**Table 2. Comparison of cost obtained by implementing 16 variants of APSO.**

| Variant. no | Minimum ($) | Maximum ($) | Average ($) | Std. deviation ($) |
|---|---|---|---|---|
| 1 | 922329.424646790 | 922339.476833197 | 922332.983292443 | 2.139597945750570 |
| 2 | 922329.020515838 | 922334.796166299 | 922331.411276286 | 1.361088807162620 |
| 3 | 922328.309587081 | 922337.839284874 | 922332.355857353 | 2.032272933703020 |
| 4 | 922327.913035623 | **922480.083143976** | **922337.100608413** | 29.226889763077900 |
| 5 | 922327.709299195 | 922334.015275867 | 922331.044136979 | 1.560290880362930 |
| 6 | 922327.089041942 | 922333.380035461 | 922330.347735337 | 1.538656482837650 |
| 7 | 922326.308678703 | 922333.251583416 | 922329.538070115 | 1.654475638133620 |
| 8 | 922325.765024078 | 922332.480979352 | 922328.474060958 | 1.470186548807700 |
| 9 | 922325.698801677 | 922331.570833448 | 922328.615632545 | 1.420360604616600 |
| 10 | 922325.066543891 | 922331.353186634 | 922327.228495890 | 1.410966866808890 |
| 11 | 922324.774597344 | 922332.102952866 | 922327.486961129 | 1.543005745185750 |
| 12 | 922324.640787209 | 922475.393031006 | 922332.193065414 | **29.3062176480051** |
| 13 | 922324.623452044 | 922331.144321152 | 922327.026445323 | 1.466999848444430 |
| 14 | 922324.397350414 | 922328.697472803 | 922326.307872664 | 1.057550967469090 |
| 15 | 922324.381682261 | 922331.002347075 | 922326.720440887 | 1.247227232472900 |
| 16 | **922323.966688770** | **922328.357852968** | **922326.214371057** | **0.975084114589519** |

are higher. Table 3 shows the comparison of the cost obtained by implementing Improved PSO with APSO variant-16 for 50 trials. Tables 4 and 5 show the power flow and the minimum cost obtained by implementing APSO variant-16 on the CSTHTS problem and Tables 6 and 7 show the power flow and the minimum cost obtained by implementing Improved PSO on the CSTHTS problem. It can be seen that no constraint of the power system under consideration has been violated. Fig 2 shows the convergence behavior of Improved PSO applied to the CSTHTS problem.

Proper statistical tests are required to establish the superiority of one algorithm over the other, according to reference [24]. So, non-parametric Mann-Whitney U test and parametric Independent samples t-test have been performed using SPSS software on the data sets obtained by running APSO variant-16 and Improved PSO for 50 trials. Results of the Mann-Whitney U test have been presented in Tables 8 and 9 and results of the Independent samples t-test have been presented in Tables 10 and 11. It can be seen that APSO variant-16 is statistically different and better than Improved PSO thus superiority of APSO variant-16 over Improved PSO has been established statistically. Table 12 shows the comparison of results achieved by

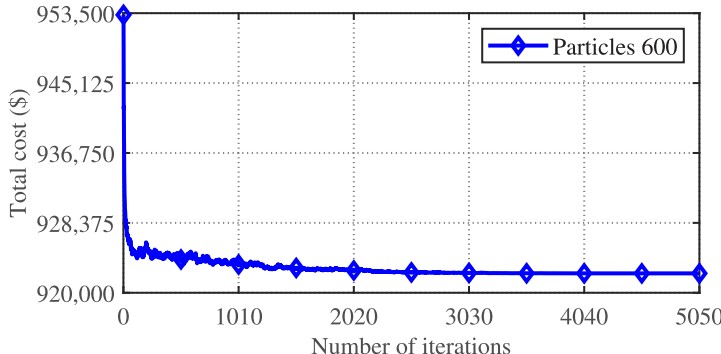

**Fig 1. Convergence characteristics of APSO variant-16 for the CSTHTS problem.**

**Table 3. Comparison of cost obtained from APSO variant-16 and Improved PSO.**

| Algorithm applied | Minimum ($) | Maximum (S) | Average ($) | Standard. deviation ($) |
|---|---|---|---|---|
| Improved PSO | 922320.6528 | 923506.5519 | 922623.8742 | 306.8276 |
| APSO variant-16 | 922323.9667 | 922328.3579 | 922326.2144 | 0.9751 |

implementing APSO variant-16 and Improved PSO on the CSTHTS problem with the reported results in the literature presented in reference [24].

## Discussion

Sixteen variants of APSO and one variant of PSO (Improved PSO) have been applied to solve the CSTHTS problem. All of them have been run for 50 trials and give better results than the so-far best result of the CSTHTS problem which is 922335.6037 $, as found in literature, according to reference [24]. It can be noticed that the APSO variant-16 is the same as that adopted by the reference [24]. However, it was observed that by selecting the maximum and minimum values of both $\alpha$ and $\beta$ as 0.81 and 0.62 respectively, the results were significantly improved, as presented in Table 2.

The minimum cost achieved by the sixteen variants and their standard deviations is improved as compared to the improved APSO-3 results previously achieved and reported in reference [24]. It can be observed that PSO in its improved version has achieved the so far best result for the CSTHTS problem under consideration in this article, even better than APSO variant-16. However, it can be observed in Table 3, that the standard deviation achieved by APSO variant-16 is far better than Improved PSO.

Tables 8 and 9 give the Mann Whitney U test and Tables 10 and 11 give the Independent samples t-test for the CSTHTS problem, between APSO variant-16 and Improved PSO. Although, Improved PSO has given the nearest approximation to the global optimum solution, however, the sixteen variants of APSO discussed are statistically far superior as compared to Improved PSO, as their standard deviations are lesser, and their minimum is very near to the best value achieved by Improved PSO. Also, the highest values of best cost achieved by sixteen variants of APSO are far lesser than the highest value of best cost achieved by Improved PSO. So, it can be concluded that sixteen variants of APSO are statistically different and superior in performance for the greatest number of trials as compared to Improved PSO.

Fig 3 gives the comparison of the convergence behavior of the best trial of the four meta-heuristic algorithms applied to the CSTHTS problem. Improved PSO and two variants of APSO (APSO variant-16 and the variant applied in reference [24]) give almost the same convergence characteristics for their best trial. Simple APSO, however, shows inferior performance. Fig 4 gives the comparison of the minimum costs achieved by the APSO variant-16 and Improved PSO, for 50 trials. Though the nearest approximation to the minimum value is achieved by Improved PSO, however, it is not much different from the minimum achieved by the APSO variant-16. Moreover, the APSO variant-16 has statistically outperformed Improved PSO.

There is a flaw in this benchmark CSTHTS problem that for some combinations of discharge rates and volumes of reservoirs, the values of hydel powers come negative, according to reference [27]. Since this is a non-pumped storage hydrothermal scheduling problem, these negative values of hydel power mean nothing and therefore are fixed to zero value, according to reference [28], and the power plant may be simply shut down during these intervals and the water may be discharged out just as spillage water (spillage water does not produce hydel power). This issue of the problem has been adopted previously as it is and all the research

**Table 4. Power flow and cost optimization with APSO variant-16 of APSO algorithm (a).**

| Interval | Volume 1 (acre-ft) | Volume 2 (acre-ft) | Volume 3 (acre-ft) | Volume 4 (acre-ft) | Q1 (acre-ft/hr) | Q2 (acre-ft/hr) | Q3 (acre-ft/hr) | Q4 (acre-ft/hr) |
|---|---|---|---|---|---|---|---|---|
| 1 | 99.9798205884863 | 80.6295627173402 | 148.100000006460 | 109.799999999290 | 10.0201794115137 | 7.370437282826598 | 29.9999999993541 | 13.000000000000705 |
| 2 | 98.6704373014632 | 82.5446664558474 | 126.300000038180 | 99.199999999295 | 10.309383287023 | 6.084896261 4929 | 29.9999999968281 | 13.000000000000000 |
| 3 | 97.6137183402722 | 85.5424805443407 | 110.320179415210 | 87.799999996790 | 9.0567189611910 | 6.0021859115067 | 29.9999999998107 | 13.000000002505 |
| 4 | 96.1281719803436 | 88.5246905466874 | 100.000000088240 | 74.799999971477 | 8.4855463599285 | 6.0177899976533 | 29.9999999763793 | 13.0000000025313 |
| 5 | 93.9396975092428 | 90.5184668394184 | 100.000000000000 | 91.799999965018 | 8.1884744711009 | 6.0062237072691 | 18.1416152315082 | 13.000000000000000 |
| 6 | 92.8625466707071 | 91.2813975507861 | 100.3632666319620 | 108.7999999922190 | 8.0771508385357 | 6.2370692886323 | 18.124465639 4732 | 13.000000011111 |
| 7 | 92.6036492935517 | 90.6686485222112 | 100.6608003372080 | 125.7999999920300 | 8.2588973771554 | 6.6127490285749 | 16.9087307635081 | 13.0000000000000 |
| 8 | 93.1314608617446 | 90.4166698784866 | 100.827763519831 0 | 142.799999684090 | 8.4721884318071 | 7.2519786437246 | 15.9164113631816 | 13.00000000000000 |
| 9 | 94.5044900876941 | 90.6147973640683 | 101.2420268752360 | 147.9416151948800 | 8.6269707740506 | 7.8018725144182 | 15.081703310 3828 | 13.000000050375 |
| 10 | 96.7989686212378 | 91.5450872079506 | 102.2239787014280 | 153.0660808129550 | 8.7055214664562 | 8.069710156117 7 | 15.1029856341898 | 13.000000213980 |
| 11 | 100.1438193392990 | 92.4209781019504 | 103.6949710984180 | 156.9748115700940 | 8.6551492819389 | 8.1241091060003 | 15.407957020785 5 | 13.000000063691 |
| 12 | 101.4889700720650 | 91.9743682998747 | 106.5173256713770 | 159.6880709909370 | 8.6548492672344 | 8.446609802075 7 | 15.6850394079157 | 13.203151942338 0 |
| 13 | 103.9944604714250 | 91.4543293736483 | 111.1297033635380 | 160.000000000000 | 8.4945090600639 1 | 8.5200389262263 | 16.1124817458954 | 14.7697743013202 |
| 14 | 107.5041088051440 | 91.7773934899289 | 114.3983015499160 | 160.000000000000 | 8.4903516662816 | 8.6769358837194 | 16.5103601868567 | 15.1029856341898 |
| 15 | 110.2065751941360 | 91.9519804025535 | 117.3887754011060 | 160.000000000000 | 8.2975336110082 | 8.8254130873753 | 16.9506455515242 | 15.4079570207855 |
| 16 | 112.1422050263140 | 90.9787414809159 | 119.0504355642790 | 160.000000000000 | 8.0643701678218 | 8.9732389216377 | 17.3487304293351 | 15.6850394079157 |
| 17 | 113.1596182176280 | 88.6572608772882 | 121.2477916207790 | 160.000000000000 | 7.9825868086858 | 9.3214806036276 | 16.7771134382281 | 16.1124817458954 |
| 18 | 113.3677193482690 | 85.0107780805137 | 124.3467785141440 | 160.000000000000 | 7.7918988693586 | 9.6464827967745 | 15.7907963618320 | 16.5103601868567 |
| 19 | 112.6684369905360 | 81.7410914816218 | 127.5881553774080 | 160.000000000000 | 7.6993723577339 | 10.269686598 8919 | 14.7144488670595 | 16.9506455515242 |
| 20 | 111.0413044194700 | 78.7998570774155 | 132.1270980523210 | 159.9999999867630 | 7.6270425710656 | 10.941234404206 3 | 13.5744367980732 | 17.348730442572 0 |
| 21 | 110.5291944346650 | 76.0853284538525 | 141.4729531936820 | 158.5241252806990 | 7.5121099848047 | 11.7145286235630 | 10.000000013476 | 18.252981442926 |
| 22 | 111.0667497180990 | 75.5417249217951 | 151.3696823636390 | 154.4941924250280 | 7.4624447165661 | 9.5436035320574 | 10.0000000000000 | 19.8207292175031 |
| 23 | 115.0001417619580 | 73.1958133379682 | 160.8230267509600 | 148.2670505986000 | 5.0666079561408 | 10.3459115838269 | 10.000000016902 | 20.941590634872 |
| 24 | 120.000000000000 | 70.000000000000 | 170.000000000000 | 140.000000000000 | 5.0001417619583 | 11.1958133337968 1 | 10.0000000910893 | 21.8414873966730 |

**Table 5. Power flow and cost optimization with APSO variant-16 of APSO algorithm (b).**

| Interval | Power Demand (MW) | P Hydel 1 (MW) | P Hydel 2 (MW) | P Hydel 3 (MW) | P Hydel 4 (MW) | P Thermal (MW) | Individual Cost ($) | Total Cost ($) |
|---|---|---|---|---|---|---|---|---|
| 1 | 1370 | 86.0853757506495 | 58.549473452838 | 0.0000000000000 | 200.0936800005480 | 1025.2714707959600 | 26787.5754169388000 | 922323.96668877 |
| 2 | 1390 | 86.8845541515415 | 51.0792936335313 | 0.0000000000000 | 187.7552799999160 | 1064.2808722150100 | 27699.5802964537000 | |
| 3 | 1360 | 80.4717503929585 | 52.1628874276380 | 0.0000000000000 | 173.7332800016760 | 1053.6320821777300 | 27450.0171070007000 | |
| 4 | 1290 | 76.7893045427112 | 53.8673748783840 | 0.0000000000000 | 156.7916800163200 | 1002.5516405625800 | 26259.2110827911000 | |
| 5 | 1290 | 74.2820885406850 | 54.8085015157776 | 25.4416841349794 | 178.7420799956610 | 956.7256458128970 | 25199.7803223198000 | |
| 6 | 1410 | 73.2302898980260 | 56.8533933205558 | 25.6854528727951 | 198.9584800010070 | 1055.2723839076200 | 27488.4293795024000 | |
| 7 | 1650 | 74.2116065290061 | 59.175160134916 | 30.2064010144491 | 217.4408799917410 | 1268.9659523303100 | 32584.6954610891000 | |
| 8 | 2000 | 75.6356975786274 | 63.3262421194186 | 33.1973328487007 | 234.1892799704880 | 1593.651447482700 | 40677.5576637969000 | |
| 9 | 2240 | 77.0134163339801 | 66.9182039116305 | 35.3718647227097 | 238.9132683446640 | 1821.7832466870200 | 46616.0267322097000 | |
| 10 | 2320 | 78.2705183505673 | 69.0467021311822 | 35.7543917845499 | 243.4636756652250 | 1893.4647120684800 | 48524.9397034118000 | |
| 11 | 2230 | 79.0999504693796 | 69.8346297868349 | 35.7184078137597 | 246.8286131844660 | 1798.5183987455600 | 46000.8901171673000 | |
| 12 | 2310 | 79.5189768384181 | 71.5059441526975 | 36.2828007362804 | 251.1803291395480 | 1871.5119491330600 | 47938.1433748503000 | |
| 13 | 2230 | 79.3133967113409 | 71.6533255425818 | 37.1647229335630 | 266.5569329974850 | 1775.3116218150300 | 45389.4458479515000 | |
| 14 | 2200 | 80.2235349500763 | 72.7231931348842 | 37.4518948811866 | 269.5756425477280 | 1740.0257344861200 | 44463.8732154815000 | |
| 15 | 2130 | 79.6668750415002 | 73.6523535091472 | 37.4042796054181 | 272.2781793589340 | 1666.9983124850000 | 42564.1343473677000 | |
| 16 | 2270 | 78.5693945571167 | 73.9429141760311 | 36.8406475332985 | 274.6835789723760 | 1605.9634647611800 | 40992.7358237101000 | |
| 17 | 2130 | 78.2241914264072 | 74.5121573809510 | 39.4759369074557 | 278.3009245011420 | 1659.4867897840400 | 42369.9391747891000 | |
| 18 | 2140 | 76.9712741584860 | 74.0309800415974 | 43.1854243363818 | 281.5662806382940 | 1664.2460408252400 | 42492.9537526496000 | |
| 19 | 2240 | 76.2063050488255 | 74.9723348567708 | 46.3859241966401 | 285.0652672737670 | 1757.3701686240000 | 44918.2070567198000 | |
| 20 | 2280 | 75.3962360406965 | 75.9552265297222 | 49.2273071246866 | 288.1254228249670 | 1791.2958074799300 | 45810.3608434050000 | |
| 21 | 2240 | 74.4950924410213 | 77.0698896459225 | 50.5929833465604 | 293.2692433575980 | 1744.5727912089000 | 44582.8660388636000 | |
| 22 | 2120 | 74.2499560681120 | 67.4455847273419 | 52.7846316487206 | 299.2485385274870 | 1626.2712890283400 | 41513.9253603799000 | |
| 23 | 1850 | 55.3192724051720 | 69.5466741372173 | 54.5854149677788 | 298.620492199740 | 1371.9281462903600 | 35105.3940859423000 | |
| 24 | 1590 | 55.0213325539675 | 70.7119598259763 | 56.0600001712479 | 293.2559686861480 | 1114.9507387626600 | 28893.2844839779000 | |

**Table 6. Power flow and cost optimization with Improved PSO algorithm (a).**

| Interval | Volume 1(acre-ft) | Volume 2(acre-ft) | Volume 3(acre-ft) | Volume 4(acre-ft) | Q1 (acre-ft/hr) | Q2 (acre-ft/hr) | Q3 (acre-ft/hr) | Q4 (acre-ft/hr) |
|---|---|---|---|---|---|---|---|---|
| 1 | 99.9745088497391 | 80.7031236589509 | 148.1000043665080 | 109.8000000000000 | 10.0254911502609 | 7.2968763410491 | 29.9999956334919 | 13.0000000000000 |
| 2 | 98.5964439895096 | 82.5164518987279 | 126.3000045939140 | 99.2000000000000 | 10.3780648602295 | 6.1866717602231 | 29.9999997725943 | 13.0000000000000 |
| 3 | 97.5411162975255 | 85.5164511246727 | 110.3254958510000 | 87.8000000000000 | 9.0553276919841 | 6.0000007740552 | 29.9999998931746 | 13.0000000000000 |
| 4 | 96.0637321869161 | 88.5164511246727 | 100.0004370522790 | 74.7999297414654 | 8.4773841106095 | 6.0000000000000 | 30.0000000000000 | 13.0000702585346 |
| 5 | 93.8481458448060 | 90.5085437594022 | 100.0004292030810 | 91.7999253749573 | 8.2155863421101 | 6.0079073652705 | 18.2420073014046 | 13.0000000000000 |
| 6 | 92.7876266221728 | 91.3398603745664 | 100.5817730052300 | 108.7999251475520 | 8.0605192226333 | 6.1686833848358 | 17.8960410825158 | 13.0000000000000 |
| 7 | 92.6429440416374 | 90.6759887210104 | 100.9634691240370 | 125.7999250407260 | 8.1446825805354 | 6.6638716535560 | 16.8338902233027 | 13.0000000000000 |
| 8 | 93.2111582099707 | 90.4116928390378 | 101.1792366114690 | 142.7999250407260 | 8.4317858316667 | 7.2642958819727 | 15.8526591004721 | 13.0000000000000 |
| 9 | 94.6006874964342 | 90.6037883858837 | 101.4549113094520 | 148.0419323421310 | 8.6104707135365 | 7.8079044531541 | 15.0376912673881 | 13.0000000000000 |
| 10 | 96.8764511641940 | 91.5384598534448 | 102.5195601211100 | 152.9379734246470 | 8.7242363322402 | 8.0653285324389 | 15.0310086735645 | 13.0000000000000 |
| 11 | 100.2148209184630 | 92.4174209966738 | 104.0103832643630 | 156.7718636479490 | 8.6616302457307 | 8.1210388567710 | 15.3839434522568 | 13.0000000000000 |
| 12 | 101.5352415905500 | 91.9374709489402 | 106.9120589561320 | 159.4165456969020 | 8.6795793279132 | 8.4799500477337 | 15.6304650936254 | 13.2079770515191 |
| 13 | 103.9555233887080 | 91.4577169586805 | 111.4970601936320 | 160.0000000000000 | 8.5797182018415 | 8.4797539902597 | 16.1419575406691 | 14.4542369642903 |
| 14 | 107.5146790491930 | 91.7486678753866 | 114.7498700280320 | 160.0000000000000 | 8.4408443395159 | 8.7090490832939 | 16.5478083502840 | 15.0310086735645 |
| 15 | 110.2076786458700 | 91.9757396543676 | 117.8068914659690 | 159.9999979745870 | 8.3070004033227 | 8.7729282210190 | 17.0026468116381 | 15.3839454776701 |
| 16 | 112.1265085170890 | 90.9978384908390 | 119.2832381234220 | 159.9999985247360 | 8.0811701287810 | 8.9779011635286 | 17.4442516723229 | 15.6304645434761 |
| 17 | 113.1515508266880 | 88.6488527366307 | 121.5321692056730 | 159.9999986294120 | 7.9749576904011 | 9.3489857542083 | 16.7671184043654 | 16.1419574359935 |
| 18 | 113.3587456572800 | 84.9515013195247 | 124.4537119507820 | 159.9999990539580 | 7.7928051694077 | 9.6973514171060 | 15.9325556046917 | 16.5478079257377 |
| 19 | 112.6596417302290 | 81.6811364967629 | 127.5702600396460 | 159.9999989792820 | 7.6991039270512 | 10.2703648227618 | 14.8363107650656 | 17.0026468863141 |
| 20 | 111.0347818294700 | 78.6981082898714 | 132.0819017282930 | 159.9999995170430 | 7.6248599007593 | 10.9830282068915 | 13.6301492349684 | 17.4442511345622 |
| 21 | 110.5077676239340 | 76.0278333444615 | 141.4783570724500 | 158.2907863678050 | 7.5270142055359 | 11.6702749454099 | 10.0000000000000 | 18.4763315536031 |
| 22 | 111.0616665067920 | 75.4308627575316 | 151.3735816678260 | 154.4384730715240 | 7.4461011171413 | 9.5969705869300 | 10.0000001281457 | 19.7848689009728 |
| 23 | 115.0036822734890 | 73.1732607062298 | 160.8836240802530 | 148.3609128012180 | 5.0579842333037 | 10.2576020513018 | 10.0000000000000 | 20.9138710353714 |
| 24 | 120.0000000000000 | 70.0000000000000 | 170.0000000000000 | 140.0000000000000 | 5.0036822734888 | 11.1732607062298 | 10.0000001428044 | 21.9910620361864 |

**Table 7. Power flow and cost optimization with Improved PSO algorithm (b).**

| Interval | Power Demand (MW) | P Hydel 1 (MW) | P Hydel 2 (MW) | P Hydel 3 (MW) | P Hydel 4 (MW) | P Thermal (MW) | Individual Cost ($) | Total Cost ($) |
|---|---|---|---|---|---|---|---|---|
| 1 | 1370 | 86.1077873385179 | 58.1297989420904 | 0.0000000000000 | 200.0936800000000 | 1025.6687337193900 | 26796.8323900714000 | 922320.652836039 |
| 2 | 1390 | 87.1495933890909 | 51.7813383509611 | 0.0000000000000 | 187.7552800000000 | 1063.3137882599500 | 27676.8971591984000 | |
| 3 | 1360 | 80.4387903492418 | 52.1329867913553 | 0.0000000000000 | 173.7332800000000 | 1053.6949428594000 | 27451.4889681155000 | |
| 4 | 1290 | 76.7199267139793 | 53.7345864045211 | 0.0000000000000 | 156.7921449271490 | 1002.7533419543500 | 26263.8926951248000 | |
| 5 | 1290 | 74.4100674719813 | 54.8156928091166 | 25.0388095941166 | 178.7419874500050 | 956.9934426747200 | 25205.9469979995000 | |
| 6 | 1410 | 73.1032388209212 | 56.3939563852472 | 26.6812238671385 | 198.9583948029260 | 1054.8631861237700 | 27478.8458564546000 | |
| 7 | 1650 | 73.5534344584183 | 59.5305097675237 | 30.5870622563970 | 217.4408023271840 | 1268.8881911904800 | 32582.8077543424000 | |
| 8 | 2000 | 75.4351471670771 | 63.4037048284998 | 33.5220235927895 | 234.1892099730300 | 1593.4491443860000 | 40672.4036168700000 | |
| 9 | 2240 | 76.9561145614169 | 66.9495994426618 | 35.5579640224001 | 239.0038596299800 | 1821.5324623435400 | 46609.3842997386000 | |
| 10 | 2320 | 78.4019986770961 | 69.0167657419542 | 36.0341364793684 | 243.3518392578280 | 1893.1952598437500 | 48517.7255727898000 | |
| 11 | 2230 | 79.1596974330591 | 69.8143106312195 | 35.9098871814732 | 246.6561560985240 | 1798.4599486557200 | 45999.3473880274000 | |
| 12 | 2310 | 79.6757631631658 | 71.6797596511249 | 36.5826502459366 | 251.0012875673610 | 1871.0605393724100 | 47926.0974399435000 | |
| 13 | 2230 | 79.8092153559049 | 71.4226200593951 | 37.2416918776032 | 263.6348816576430 | 1777.8915910494500 | 45457.3155671982000 | |
| 14 | 2200 | 79.9234902261820 | 72.8895694388376 | 37.4923578125550 | 268.9294001588230 | 1740.7651823636000 | 44483.2183416399000 | |
| 15 | 2130 | 79.7270681324400 | 73.3711821253855 | 37.4182118529212 | 272.0674880148520 | 1667.4160498744000 | 42574.9407243460000 | |
| 16 | 2270 | 78.6768437868693 | 73.9789017034147 | 36.6243000373735259 | 274.2135683338530 | 1606.5063858023400 | 41006.6481426523000 | |
| 17 | 2130 | 78.1716099838292 | 74.6511729493625 | 39.6172771625416 | 278.5461941138890 | 1659.0137457903800 | 42357.7171366181000 | |
| 18 | 2140 | 76.9758582374261 | 74.2480007345386 | 42.9028563025568 | 281.8685567071280 | 1664.0047280183500 | 42486.7142476872000 | |
| 19 | 2240 | 76.2028425181839 | 74.9370371715421 | 46.1872514124507 | 285.4705896451340 | 1757.2022792526900 | 44913.8034620733000 | |
| 20 | 2280 | 75.3798414894861 | 76.0581229919429 | 49.1677147151055 | 288.8450920557300 | 1790.5492287477400 | 45790.6782730947000 | |
| 21 | 2240 | 74.5951828596326 | 76.8688068109337 | 50.5942655481191 | 294.5783507648080 | 1743.3633940165100 | 44551.2090123104000 | |
| 22 | 2120 | 74.1334320008320 | 67.6312071285805 | 52.7854335951837 | 298.9788318003670 | 1626.4710954750400 | 41519.0614819522000 | |
| 23 | 1850 | 55.2402577055474 | 69.1406452468902 | 54.5960358198607 | 298.5857539846320 | 1372.4373072430700 | 35117.9646236921000 | |
| 24 | 1590 | 55.0546076759563 | 70.6253740079106 | 56.0600002684722 | 293.8829720647810 | 1114.3770459828800 | 28879.7116840984000 | |

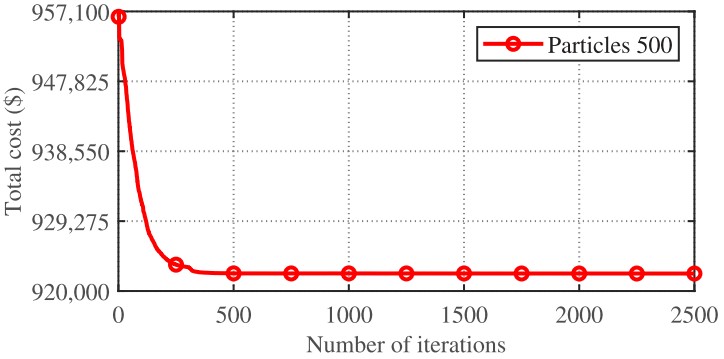

**Fig 2. Convergence characteristics of Improved PSO for the CSTHTS problem.**

**Table 8. Non-parametric Mann Whitney U test for the CSTHTS problem.**

| Test Statistics | |
|---|---|
| **Mann-Whitney U** | 750.000 |
| **Wilcoxon W** | 2025.000 |
| **Z** | -3.447 |
| **Asymp. Sig. (2-tailed)** | 0.001 |

**Table 9. Rank statistics of non-parametric Mann Whitney U test for the CSTHTS problem.**

| Ranks | | | |
|---|---|---|---|
| Group | N | Mean Rank | Sum of Ranks |
| **APSO variant-16** | 50 | 40.50 | 2025.00 |
| **Improved PSO** | 50 | 60.50 | 3025.00 |
| **Total** | 100 | | |

**Table 10. Group statistics of independent sample t-test for the CSTHTS problem.**

| Group Statistics | | | | |
|---|---|---|---|---|
| Group | N | Mean | Std. Deviation | Std. Error Mean |
| **APSO variant-16** | 50 | 922326.2144 | 0.9750841146 | 0.1378977179 |
| **Improved PSO** | 50 | 922623.8742 | 306.8276371 | 43.39198057 |

articles related to this CSTHTS problem, as mentioned in the literature, till date adopt the same mathematical model, and thus the optimal scheduling is taken as it is.

## Conclusion

An important case of the CSTHTS problem has been solved by implementing sixteen newly proposed variants of the APSO algorithm and one newly proposed variant of the PSO algorithm. The results achieved are better than the reported so far best results achieved by the other algorithms, as found in the literature, according to reference [24]. The proposed algorithms, not only helped to achieve the so far best approximation of the global optimum solution, but also they achieved it persistently in every repeated trial, i.e., by maintaining low standard deviation, and in very economical computation time.

**Table 11. Independent sample t-test for equality of means for the CSTHTS problem.**

| Independent samples test | Levene's test for equality of variances | | t-test for equality of means | | | | | | |
|---|---|---|---|---|---|---|---|---|---|
| | F | Sig. | t | df | Sig. (2-tailed) | Means difference | Std. error difference | 95% confidence interval of the difference | |
| | | | | | | | | Lower | Upper |
| Equal variances assumed | 83.742 | .000 | -6.860 | 98 | .000 | -297.659782 | 43.39219969 | -383.770190 | -211.549373 |
| Equal variances not assumed | | | -6.860 | 49.001 | .000 | -297.659782 | 43.39219969 | -384.859627 | -210.459936 |

It can be concluded that APSO in its canonical as well as in improved form gives good approximates to the global best solution and is easy to program. PSO also gives good approximates to the global optimum solution but on the statistical basis, it has been established that statistically, the improved variants of APSO outperformed the improved PSO variant on the CSTHTS problem, though, the improved PSO helped to achieve the so far best approximate of the global optimum solution.

**Table 12. Comparison of cost obtained by implementation of different algorithms on the CSTHTS problem.**

| Algorithm | Minimum cost ($) | Average cost ($) | Maximum cost ($) | Computation time (sec) |
|---|---|---|---|---|
| FEP [7] | 930267.92 | 930897.44 | 931396.81 | NA |
| CEP [7] | 930166.25 | 930373.23 | 930927.01 | NA |
| IFEP [7] | 930129.82 | 930290.13 | 930881.92 | NA |
| GA [2] | 932734 | 936969 | 939734 | NA |
| IPSO [8] | 922553.49 | NA | NA | NA |
| APSO [9] | 926151.54 | NA | NA | NA |
| MAPSO [9] | 922421.66 | 922544 | 923508 | NA |
| DE [10] | 923991.08 | NA | NA | NA |
| BCGA [11] | 926922.71 | 927815.35 | 929451.09 | NA |
| RCGA [11] | 925940.03 | 926120.26 | 926538.81 | NA |
| EGA [12] | 934727 | 936058 | 937339 | NA |
| PSO [12] | 928878 | 933085 | 938012 | NA |
| EPSO [12] | 922904 | 923527 | 924808 | NA |
| MDE [13] | 922555.44 | NA | NA | NA |
| TLBO [14] | 922373.39 | 922462.24 | 922873.81 | NA |
| RCGA-AFSA [15] | 922340 | 922362 | 922346 | NA |
| MFO [16] | 924455 | 925431 | 924836 | NA |
| GWO [16] | 924259 | 925210 | 924784 | NA |
| PSO-ALNS [16] | 923542 | 924025 | 923755 | NA |
| CGWO-DA [16] | 923259 | 923711 | 923444 | 67 |
| SPPSO [17] | 922336.31 | NA | NA | 16 |
| Canonical APSO [24] | 922615.3048 | 923322.9877 | 924967.5195 | 80 |
| Improved APSO [24] | 922335.6037 | 922351.7587 | 922443.6 | 100 |
| **Proposed APSO variant-16** | 922323.9667 | 922326.2144 | 922328.3579 | 45 |
| **Proposed Improved PSO** | 922320.6528 | 922623.8742 | 923506.5519 | 34 |

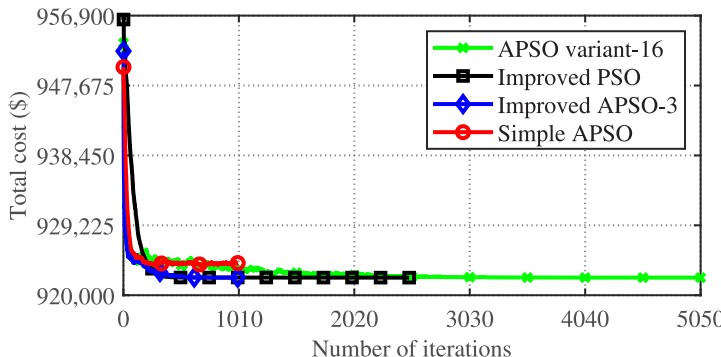

**Fig 3. Comparison of the convergence behavior of the best trials of four metaheuristic algorithms on the CSTHTS problem.**

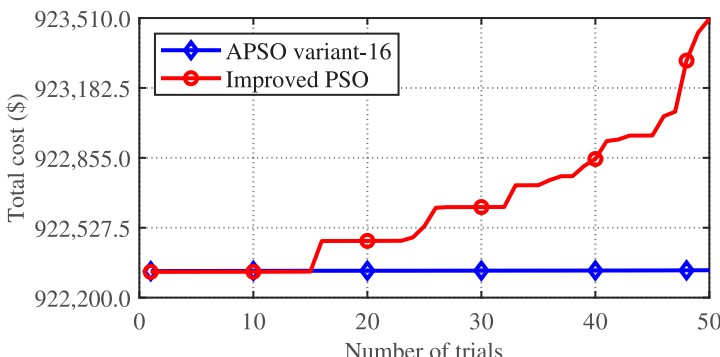

**Fig 4. Comparison of minimum costs achieved by APSO variant-16 and Improved PSO for 50 trials for the CSTHTS problem.**

Furthermore, it has been established that true statistical hypothesis testing is mandatory to validate the superiority of one type of metaheuristic algorithm over the other type of metaheuristic algorithm for a given short-term hydrothermal scheduling problem.

## Author Contributions

**Conceptualization:** Muhammad Ahmad Iqbal, Muhammad Salman Fakhar, Rehan Naeem.

**Formal analysis:** Muhammad Ahmad Iqbal, Muhammad Salman Fakhar, Syed Abdul Rahman Kashif, Rehan Naeem, Akhtar Rasool.

**Investigation:** Muhammad Ahmad Iqbal, Muhammad Salman Fakhar.

**Methodology:** Muhammad Ahmad Iqbal, Muhammad Salman Fakhar.

**Project administration:** Muhammad Ahmad Iqbal, Muhammad Salman Fakhar.

**Supervision:** Muhammad Ahmad Iqbal, Muhammad Salman Fakhar.

**Validation:** Muhammad Ahmad Iqbal, Muhammad Salman Fakhar, Syed Abdul Rahman Kashif, Rehan Naeem, Akhtar Rasool.

**Writing – original draft:** Muhammad Ahmad Iqbal, Muhammad Salman Fakhar.

**Writing – review & editing:** Syed Abdul Rahman Kashif, Rehan Naeem, Akhtar Rasool.

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
