## [Decision Letter · Decision Letter 0]

12 Oct 2021

PONE-D-21-29734Impact of parameter control on the performance of APSO and PSO algorithms for the CSTHTS problem: An improvement in algorithmic structure and resultsPLOS ONE

Dear Dr. Fakhar,

Thank you for submitting your manuscript to PLOS ONE. After careful consideration, we feel that it has merit but does not fully meet PLOS ONE’s publication criteria as it currently stands. Therefore, we invite you to submit a revised version of the manuscript that addresses the points raised during the review process.

We look forward to receiving your revised manuscript.

Kind regards,

Yang Li

Academic Editor

PLOS ONE

Journal Requirements:

"No"

"No competing interests."

Additional Editor Comments:

The manuscript was evaluated by three reviewers and detailed comments are provided. The team agrees that the topic is relevant and of high contemporary interest, and that this work is basically technically sound. However, there are still some major issues that are needed to be addressed before possible publication. Taking into account these factors, my recommendation is for the paper to be "Major Revision".

Reviewers' comments:

Reviewer's Responses to Questions

**Comments to the Author**

1. Is the manuscript technically sound, and do the data support the conclusions?

Reviewer #1: Yes

Reviewer #2: Yes

Reviewer #3: Partly

2. Has the statistical analysis been performed appropriately and rigorously? 

Reviewer #1: Yes

Reviewer #2: Yes

Reviewer #3: Yes

3. Have the authors made all data underlying the findings in their manuscript fully available?

Reviewer #1: Yes

Reviewer #2: Yes

Reviewer #3: Yes

4. Is the manuscript presented in an intelligible fashion and written in standard English?

Reviewer #1: Yes

Reviewer #2: Yes

Reviewer #3: Yes

5. Review Comments to the Author

Reviewer #1: In this paper, the authors analyze the impact of parameter control on the performance of APSO and PSO algorithms for the CSTHTS problem. Here, there are some concerns of this reviewer:

1 Introduction can be improved by having four clear and concise subsections on motivation of your research; background and related works; list of key contributions/future trends; and organization of the paper.

2 The novelty and main contributions of this paper should be further summarized and clearly demonstrated. This reviewer suggests the authors exactly mention what is new compared with existing approaches and why the proposed approach is needed to be used instead of the existing methods. Especially, the comparison this paper with authors’ previous works [R1, R2] must be performed to highlight the contributions of this work.

[R1] Fakhar MS, Kashif SAR, Ain NU, Hussain HZ, Rasool A, Sajjad IA. Statistical performances evaluation of APSO and improved APSO for short term hydrothermal scheduling problem. Applied Sciences. 2019;9(12):2440.

[R2] Liaquat S, Fakhar MS, Kashif SAR, Rasool A, Saleem O, Padmanaban S. Performance Analysis of APSO and Firefly Algorithm for Short Term Optimal Scheduling of Multi-Generation Hybrid Energy System. IEEE Access. 2020;8:177549–177569.

3 The authors should introduce more about the references, not only mentioning them.

4 In this study, how to handle the problem of premature convergence in the improved variant of the Accelerated PSO (APSO) algorithm? Use of mutation strategy together with dynamic monitoring mechanism of diversity has been proven as an effective method reported in literature like [doi.org/10.1155/2015/529724]. Please discuss this.

5 The proposed method might be sensitive to the values of its main controlling parameters. How did you determine these optimal parameters?

6 The Conclusions can be improved. This reviewer strongly suggests the authors clearly explain what the significant findings are and why your paper is really important. Some of the most important quantitative results should be reported to better demonstrate the findings of the carried-out work.

7 The nomenclature should be included to improve the readability of this paper.

8 All references, including article titles, should be in a uniform format. Please double-check it.

Reviewer #2: The manuscript entitled,” Impact of parameter control on the performance of APSO and PSO algorithms for the CSTHTS problem: An improvement in algorithmic structure and results” describes a study on parametric sensitivities of evolutionary algorithms which is interesting, however, apparently the study seems very close to the authors earlier works:

Fakhar, M. S., Kashif, S. A. R., Ain, N. U., Hussain, H. Z., Rasool, A., & Sajjad, I. A. (2019). Statistical performances evaluation of APSO and improved APSO for short term hydrothermal scheduling problem. Applied Sciences, 9(12), 2440.

Fakhar, M. S., Liaquat, S., Kashif, S. A. R., Rasool, A., Khizer, M., Iqbal, M. A., ... & Padmanaban, S. (2021). Conventional and metaheuristic optimization algorithms for solving short term hydrothermal scheduling problem: A review. IEEE Access.

Fakhar, M. S., Kashif, S. A. R., Liaquat, S., Rasool, A., Padmanaban, S., Iqbal, M. A., ... & Khan, B. (2021). Implementation of APSO and Improved APSO on Non-Cascaded and Cascaded Short Term Hydrothermal Scheduling. IEEE Access.

Therefore, I would just as below:

1. Please modify the presented work in such a way that would make it exclusive.

2. Alternatively, the present study may be published as an addendum to the above-mentioned works.

3. Please do not consider the comments as a denial of your work, the comments are only suggestions.

Based on all the above comments, I would suggest a major revision and reconsideration of the presented work.

Reviewer #3: The authors presented accelerated particle swarm optimization (APSO) algorithm for solving cascaded hydrothermal scheduling problem. Although the research article has presented valuable contributions but require some minor changes in the manuscript.

1. It is suggested to organize as follows the complete manuscript (1) introduction (2) formulation (3) proposed technique (4) simulation results (5) conclusions.

2. What is the feasibility analysis of all the variants in terms cost calculations of the given problem? It is suggested to mention a justification of variants as mentioned in table 2 and provide the comment on each variant.

3. What is the significance of inertia weight parameters? Is there any influence on the variation of alpha and beta parameters?

4. On what analysis basis the improved PSO is better as compared to other variants of PSO?

5. Check the standard deviation of the proposed techniques

6. Why the hydro generation is zero in table 5 for third hydro plant even the discharge shown in table 4? Is there any technical reason or the calculations mismatch?

6. PLOS authors have the option to publish the peer review history of their article (what does this mean?). If published, this will include your full peer review and any attached files.

Reviewer #1: No

Reviewer #2: No

Reviewer #3: **Yes: **Dr Gouthamkumar Nadakuditi

---

## [Author Response · Author response to Decision Letter 0]

10 Nov 2021

We have uploaded the word file having responses to the reviewers and the corresponding actions taken. We have also upload the highlighted manuscript with track changes, along with the clean copy of manuscript.

---

## [Decision Letter · Decision Letter 1]

6 Dec 2021

Impact of parameter control on the performance of APSO and PSO algorithms for the CSTHTS problem: An improvement in algorithmic structure and results

PONE-D-21-29734R1

Dear Dr. Fakhar,

We’re pleased to inform you that your manuscript has been judged scientifically suitable for publication and will be formally accepted for publication once it meets all outstanding technical requirements.

Kind regards,

Yang Li

Academic Editor

PLOS ONE

Additional Editor Comments (optional):

Reviewers' comments:

Reviewer's Responses to Questions

**Comments to the Author**

1. If the authors have adequately addressed your comments raised in a previous round of review and you feel that this manuscript is now acceptable for publication, you may indicate that here to bypass the “Comments to the Author” section, enter your conflict of interest statement in the “Confidential to Editor” section, and submit your "Accept" recommendation.

Reviewer #1: All comments have been addressed

Reviewer #2: (No Response)

Reviewer #3: All comments have been addressed

2. Is the manuscript technically sound, and do the data support the conclusions?

Reviewer #1: Yes

Reviewer #2: Yes

Reviewer #3: Partly

3. Has the statistical analysis been performed appropriately and rigorously? 

Reviewer #1: Yes

Reviewer #2: Yes

Reviewer #3: Yes

4. Have the authors made all data underlying the findings in their manuscript fully available?

Reviewer #1: Yes

Reviewer #2: Yes

Reviewer #3: Yes

5. Is the manuscript presented in an intelligible fashion and written in standard English?

Reviewer #1: Yes

Reviewer #2: Yes

Reviewer #3: Yes

6. Review Comments to the Author

Reviewer #1: Authors have fully considered my comments on the previous version of the paper. I'm satisfied with the modifications made in the revised version. I think this paper deserves to be published in its current form.

Reviewer #2: (No Response)

Reviewer #3: The authors presented accelerated particle swarm optimization (APSO) algorithm for solving cascaded hydrothermal scheduling problem and the revised article in the present form of manuscript is recommended.

7. PLOS authors have the option to publish the peer review history of their article (what does this mean?). If published, this will include your full peer review and any attached files.

Reviewer #1: No

Reviewer #2: No

Reviewer #3: **Yes: **Dr Gouthamkumar Nadakuditi

---

## [Editor Report · Acceptance letter]

7 Dec 2021

PONE-D-21-29734R1 

Impact of parameter control on the performance of APSO and PSO algorithms for the CSTHTS problem: An improvement in algorithmic structure and results 

Dear Dr. Fakhar:

I'm pleased to inform you that your manuscript has been deemed suitable for publication in PLOS ONE. Congratulations! Your manuscript is now with our production department. 

Kind regards, 

on behalf of

Professor Yang Li 

Academic Editor

PLOS ONE